# To Get Vaccinated or Not? The Vaccination Decision-Making by Healthcare Professionals Working in Haematology: A Qualitative Study

**DOI:** 10.3390/ijerph20105901

**Published:** 2023-05-21

**Authors:** Gian Luca Tunisi, Elisa Ambrosi, Giorgia Zulianello, Elisabetta Allegrini, Domenico Provenzano, Tiziana Rizzello, Federica Canzan

**Affiliations:** 1Servizio Professioni sanitarie Azienda Ospedaliera, Universitaria Integrata di Verona, 37100 Verona, Italy; ginaluca.tunisi@aovr.veneto.it (G.L.T.); elisabetta.allegrini@aovr.veneto.it (E.A.); domenico.provenzano@aovr.veneto.it (D.P.); 2Dipartimento di Diagnostica e Sanità Pubblica, Università degli studi di Verona, 37100 Verona, Italy; elisa.ambrosi_01@univr.it; 3Department of General and Pancreatic Surgery, The Pancreas Institute, Azienda Ospedaliera Universitaria Integrata di Verona, 37100 Verona, Italy; giorgia.zulianello@aovr.veneto.it; 4Unità di Ematologia, “Pia Fondazione Card G. Panico” Ospedale Tricase, 73100 Lecce, Italy

**Keywords:** haematological malignancies, healthcare workers’ attitude, qualitative study, vaccine hesitancy, vaccine attitude

## Abstract

Haematological patients are more susceptible to infections. Vaccination has always been the most effective primary prevention strategy, even during the COVID-19 pandemic. However, the efficacy of vaccines for some haematological patients is low. Although vaccination of Healthcare Workers (HCWs) could protect patients from vaccine-preventable diseases, there is evidence of a high level of hesitation among healthcare workers in Italy. The aim of this study was to explore the attitudes towards vaccination of HCWs caring for haematology patients. Qualitative descriptive design was conducted. Twenty-one HCWs were interviewed. Content analysis was applied to the qualitative data. The following themes were generated from the analysis: “Trust”, “Decision-making process focusing on individual health”, “Decision-making process focusing on community health”, “Changing opinion”, and “Two sides of vaccination commitment”. The most hesitant HCWs were oriented towards individual health. They perceived a lack of benefit from vaccines, feared side effects, or were influenced by negative experiences of others. In contrast, community-health-oriented HCWs showed more positive attitudes towards vaccination. Some hesitant HCWs changed their opinion on vaccination because they began to reflect on the importance of vaccination for the community. The change in opinion of some HCWs interviewed provided insight into the importance of focusing organisational efforts on collective responsibility.

## 1. Introduction

The patients affected by haematological malignancies, due to their immunodeficiency condition, are more susceptible to infections [1]. In case of respiratory virus infection, these patients, especially those who have undergone bone marrow transplantation, are at risk of serious complications (e.g., pneumonia), with a mortality rate associated with the infection >50% [2,3]. In the context of the Coronavirus pandemic, several studies have shown an increased risk of serious complications and mortality associated with COVID-19 infection in patients with immunosuppression [4,5]. A study conducted in a sample of 500 Italian haematological patients showed a mortality rate four times higher than that of the general population contracting the virus [4]. These data suggest that special preventive and protective measures should be provided for patients with immunosuppression in response to the COVID-19 pandemic [6]. Vaccination has always been the most effective primary prevention strategy, even during the COVID-19 pandemic [7]. However, the effectiveness of vaccines for some haematological patients is low [1,8,9]. Several systematic reviews have shown that haematological patients, particularly those who have received B-cell depletion agents in the last 12 months, have a high post-vaccination COVID-19 nonresponse rate [9,10]. Different studies demonstrated that the vaccination of HCWs could protect patients from vaccine-preventable diseases [11,12,13,14]**,** despite this topic still being controversial [15]. Frenzel et al. [12] showed that the increased adherence to influenza vaccination among Healthcare Workers was associated with a decrease in nosocomial influenza infections in cancer patients. Therefore, the priority vaccination of Healthcare Workers (HCWs) represents a fundamental strategy to protect from a possible infection not only in the HCWs themselves [16] but also all the patients who are not able to develop an adequate humoral response to vaccination [17]. Several studies, however, have shown a high level of hesitation among healthcare professionals [18], especially in Italy [19]. The spread of greater hesitation around the world with respect to vaccinations has prompted the WHO to consider the hesitation to vaccinate as one of the ten threats to global health in 2019 [20]. Vaccine hesitancy can be defined as the delay in acceptance or the refusal of vaccination despite the availability of vaccine services. Vaccine hesitancy is complex and context specific, varying across time, place, and vaccines. It is influenced by factors such as complacency, convenience, and trust [21]. Among the barriers described by health professionals, the fear of an adverse reaction is certainly the most reported topic [22,23], and having heard of previous adverse reactions based on second- or third-hand descriptions has a negative impact [24]. Another reported barrier is the idea that one’s immune system is able to cope in a “natural” way with a possible infection without the need to resort to vaccinations. Personal protection, the reduction in the risk of contracting an infection, and the sense of responsibility towards patients are instead the reasons most reported as motivations for vaccination [24,25]. In contrast, little is known about the vaccination attitude of HCWs caring for frail haematology patients and whether this frail condition influences their motivation to vaccinate. The aim of this study was to explore the attitudes towards HCWs caring for haematology patients with a particular focus on vaccination for the SARS-CoV-2 virus.

## 2. Materials and Methods

The study was conducted in Italy in two different sites. The two centres were chosen as representative of a heterogeneous landscape characterizing the spread of COVID-19 in Italy at the time of interview. Site 1, a University Hospital in the Country’s North-East region, was in an area of moderate risk for contagion. It was subject to medium intensity preventative restrictions. However, it stood geographically close to the nation’s worst hit region, with growing concern of virus circulation. Site 2, a Tertiary Hospital located in the South, was in a low-risk area, with light restriction measures, but witnessed a surge in the number of people affected by SARS-CoV-2 [26].

A Qualitative Descriptive design [27] was conducted to investigate HCWs’ attitudes towards vaccinations. The qualitative descriptive design was based on the typical theory of naturalism. This theory sought to study a phenomenon in its natural state and in as free of an artifice as possible [27].

Purposive sampling was performed. Physicians, nurses, and nurse aides (NAs) were interviewed after signing the informed consent. Inclusion criteria for enrolment were age of 18 and older and active service in the Department of Haematology or Bone Marrow Transplant Centre for at least 1 year.

A semi-structured 30 min interview was conducted face-to-face by a single interviewer. The interviews were conducted using a format to guide the questions (Table 1).

The items recorded for each subject included age, gender, education, and years of work experience. The interviews were conducted until the subjects were saturated.

The interviews were audio recorded, with permission from the interviewees, and transcribed verbatim by 3 authors (XX.; XX.; XX.). To ensure anonymity, the participants were assigned an identification code upon transcription.

Content analysis was applied to the qualitative data, wherein each transcription was analysed independently by 2 authors (XX., XX.), and themes were generated. No software was used for data analysis. The analytical process consisted of 4 phases:Familiarization with the source material through multiple readings of the transcriptions;Identification of meaningful phrases relative to the subject’s inclinations, preconceived ideas, and intentions toward vaccines;Labelling of topical phrases;Arrangement of the labels in patterns based on similarities and definition of categories;

The analysis process was then reviewed by a third author (T.G.), and the data were tested for coherence with the labels. At this time, field notes gathered during the interviewing and analysis processes were discussed, and the authors examined how their perceptions contributed to the shape and developed their understanding of the data.

Field notes were collected during the interviews and data analysis. To increase the validity of the findings, feedback was sought from the participants, ensuring that their meanings and perspectives were represented and not limited to the researchers’ agenda. Ethical approval was obtained from both sites’ Research Ethics Committees (N).

## 3. Results

Twenty-one HCWs (6 physicians, 7 nurse, 8 nurse aides) were interviewed across the two sites from January 2021 to July 2021. The study was conducted during the first vaccination campaign for COVID-19. Most participants were female (85.7%) with an average age of 47 (min–max 26–62). In total, 92.3% of NAs and nurses had more than 10 years of work experience, while 50% of physicians had less than 5 years. The individual demographics of the HCW are given in detail in Table 2.

The intentions to be vaccinated against SARS-CoV-2 and other viruses were mixed: HCWs were more likely to take the COVID-19 vaccine rather than the flu vaccine. The differences and similarities that emerged from the interviews regarding HCWs’ attitudes between the COVID-19 vaccine and other vaccinations will be further reported for each theme/sub-theme. Nurse aides were most hesitant about vaccination, followed by nurses. The physicians interviewed, on the other hand, showed complete adherence to vaccination campaigns for COVID-19 and other infections.

Following data analysis, the attitudes of HCWs in Italy toward the vaccines were allocated to one of five themes: “Trust”, “Decision-making process focusing on individual health”, “Decision-making process focusing on community health”, “Changing opinion”, and “Two sides of vaccination commitment”. Figure 1 shows the connection between themes and sub-themes.

### 3.1. Theme 1: Trust

The most prevalent theme in interviews of HCWs who intended to get vaccinated against COVID-19 was Trust.

For most of the NAs and some nurses, trust in the novel COVID-19 vaccine was a direct consequence of trust in the National Regulatory Institutions who presided over vaccine development, safety control, and release for public use (e.g., AIFA, EMA). These NAs expressed faith that the health authorities had upheld their role in guaranteeing that shortcuts were not taken in the race to get vaccines approved.

Physicians also expressed trust in Health Institutions but felt the need to make an informed decision that took into consideration the new technologies needed to produce an mRNA vaccine, noting to have reviewed research literature to support their choice.
*“I don’t think and I never thought that the stages to develop a vaccine didn’t get observed, had it been dangerous, they wouldn’t have made it, tested it and got it approved.” **(NA)*
*“I tried to learn more about the COVID-19 vaccine, I looked it up on PubMed, I read articles, data-sheet. I looked for pre-clinical data and, although I couldn’t find them for this specific vaccine, I found some for other flu vaccines developed* via *the same method, in which they explained the pharmacodynamic and pharmacokinetic. Considering what risks, we’re facing; any vaccine is fine.”**(Physician)*

A minority of HCWs, albeit getting vaccinated and believing it might be beneficial for public health, were wary of the vaccine’s long-term effects, and perceived that clinical data were insufficient at the moment.
*“Actually, I don’t trust it … they haven’t shown us sufficient data, we don’t know how the vaccine works. I am the most doubtful about long-term side effects.”**(NA)*

Other HCWs were firmly opposed to vaccine uptake (both COVID-19 and other vaccines) because they lacked trust in clinical trials or substances administered through vaccines. Being the firsts in the country to get vaccinated left some HCWs feeling as if they were being experimented on. This perception was impacted by biases against the pharmaceutical companies developing the vaccine, supported by the belief that clinical trials had not been carried out rigorously and had been rushed.

### 3.2. Theme 2: Decision Making Process Focused on Individual Health

The theme of “Decision making process focused on individual health” contained three subthemes, including “Lack of Perceived Benefits”, “Fear of Side Effects”, and “Negative Experiences of others”. All sub-themes shared the same thought process, which was that the vaccine was a health treatment that only had an impact on individual health.

#### 3.2.1. Lack of Perceived Benefits

Some HCWs, prevalently NAs and nurses, perceived a lack of benefit from vaccination, specifically in regard to the flu vaccination. Having never been sick after contracting the flu, they did not consider themselves vulnerable, which then led them to question if vaccination was at all necessary in their case. They claimed to deem their bodies were equipped to protect themselves from illness. Many HCWs argued that the flu vaccine was not necessary in the absence of chronic diseases. Some disclosed that they could excuse the chronically ill at-risk population too from declining to uptake the flu vaccine, as it is, in their experience, a common ailment that does not manifest with severe signs and symptoms.
*“Because I have never had the flu since when I was young perhaps, it’s been at least 10 years without having the flu or even just high temperature, nothing at all, so I’m sure I won’t even pass it to the patients, because I don’t have it.”**(nurse)*

After contracting the disease, some HCWs came to question how useful the COVID-19 vaccine would be in their case, and few mentioned doubts about its benefits for healthy individuals too. In such cases, they held the immune system capable of fighting the disease on its own, with only minor inconveniences, and that enforcing standard precautions and social distancing were the most advisable means of primary prevention. Some nurses, in fact, preferred to only adopt standard precautions and cautious behaviours rather than vaccinate themselves for COVID-19.

Moreover, one nurse believed that the vaccine for COVID-19 did not help with antibodies’ production, but rather supplied receptors that helped in the early detection of the virus in case of exposure.
*“If you get the vaccine, but then you catch the virus, you can’t be sure that you are showing mild symptoms because of the vaccine or of your own immune response. See my experience with COVID.”**(NA)*
*“I sincerely prefer to take all precautions and behave correctly, as I have done so far, without… that is, I don’t go to crowded places, I don’t do anything, I do everything I can to avoid contagion.”**(Nurse)*

#### 3.2.2. Fear of Side Effects

Among the HCWs interviewed, some NAs and nurses expressed concern about the delayed side effects of vaccines. They thought the side effects were mainly caused by vaccine excipients. Despite fear, many HCWs have subjected themselves and even their children to various vaccinations.
*“With it not being 100% safe, we don’t know what long term effects it may have, that’s why in beginning I was a bit hesitant.”**(NA)*
*“However, I’ve always been a bit hesitant, also because I know that vaccines have excipients like lead in it, so they are dangerous for kids.”**(Nurse)*

#### 3.2.3. Negative Experiences of Others

Anecdotal experiences appeared to be salient factors in vaccine hesitancy. Related events of side effects from vaccines, especially in children, were trusted and unchallenged sources of information that resulted in fear of vaccination and led these healthcare workers to start changing their minds about vaccines in some cases.
*“Not in my case directly, but I met a Healthcare worker who says: << getting a vaccine, her first vaccine, my daughter became autistic.>>”**(NA)*

### 3.3. Theme 3: Decision Making Process Focused on Community Health

The theme of “Thought process focused on community health” contained three sub-themes, including “Safekeeping family members”, “Ending the pandemic”, and “Preventing Disease for Frail patients”.

#### 3.3.1. Safekeeping Family Members

Some HCWs said they felt lucky and glad to have received the vaccine before others, and they felt that they were protecting themselves and family members by getting vaccinated.
*“I agreed to all the vaccinations that were recommended to me, even the ones that weren’t mandatory, for my daughter’s benefit.”**(Physician)*

#### 3.3.2. Ending the Pandemic

Some HCWs argued that the COVID-19 vaccine could protect the entire population. They were convinced that even those who could not be vaccinated would be protected from contagion once Herd Immunity was achieved. Many Interviewees said they believed vaccines were the way to end the pandemic and were eager to get vaccinated, even though they retained doubts and concerns regarding the COVID-19 vaccine.
*“I’m fully convinced that if we reach an immunity in the population, it will be possible to protect those who can’t get vaccinated for other reasons, and it’s right to have a public health that it’s as possible.”**(Physician)*
*“I would probably get this vaccine because this pandemic is part of it, there are many risks for people, I’ve seen that it’s a very aggressive virus, but at the same time I’m not completely convinced because I’m not sure this vaccine will give total immunity.”**(Nurse)*

#### 3.3.3. Preventing Disease in Frail Patients

Many HCWs considered it necessary to undergo vaccinations to protect frail people. They thought that prevention was even more important when assisting patients with haematological diseases.
*“Clearly vaccinations are fundamental both because we’re Health professionals to prevent infecting our frail patients and also because through vaccinations we’re able to eradicate and prevent many diseases.”**(Nurse)*
*“I think vaccinations are very important, because they give an acquired immunity, so they contribute to frail people’s well-being.”**(Physician)*

### 3.4. Theme 4: Changing Opinion

Some NAs discussed how they came to change their mind and became positively inclined towards the COVID-19 vaccine uptake. Some said that they turned to prominent experts that were interviewed by media outlets, trusting their opinion because they recognised their professional expertise on virology and vaccines. For many however, the deciding factor in overcoming vaccine hesitancy was an honest exchange and dialogue with colleagues, as all interviewees attested many conversations were had at their place of work regarding the COVID-19 vaccine. One NA reported that a change of mind happened when a colleague made them realise the impact that a vaccine would have on the community, and immunocompromised patients specifically, in their case. They came around the idea of getting vaccinated when they shifted their focus from personal benefit to protecting the community.
*“Look, at first I was saying ”No, I won’t get the vaccine.” Now I changed my mind. Maybe because I heard professor ****** speak. So, if you, a professor, get vaccinated it means you have read about it, seen all the characteristics of this vaccine, so I trust you. So, I think I’ll vaccinate. Yeah, I think so.”**(NA)*
*“It made me change my mind, but always not for myself, for others... made me think that the vaccine is needed for the people around me, not so much for me… not for the work that asks me to vaccinate myself.”**(NA)*

### 3.5. Theme 5: Two Sides of Vaccination Commitment

The theme of “two sides of Vaccination Commitment” contained two subthemes, including “Lack of freedom of choice” and “Moral Requirement”.

#### 3.5.1. Lack of Freedom of Choice

Some Nurses and NAs thought it was very serious to impose vaccination on people because it harmed the sphere of personal health. They emphasised that the freedom of choice regarding vaccination was similar to other existing forms of freedom such as freedom of speech and freedom to vote. They thought that mandatory vaccination removed free will and limited freedom of choice, especially considering that they maintained their perplexities on whether the vaccine was really safe.
*“I must be free … as there is freedom of speech, freedom of vote. One has the right to say no.”**(Nurse)*
*“If it’s mandatory, you have to do it, whether you like it or not, you do it and it ends there, but, remove in this way the free will, the freedom of choice especially because it is a vaccine not totally safe.”**(NA)*

Some physicians believe that making vaccines mandatory for everyone is quite controversial, and, ideally, it would be preferable to empower to people to make this choice on their own and to consider how their choices affect public health.
*“It is difficult. OK yes, you can make [3] mandatory but then what do you do with those who don’t get it? Ideally, people should understand and there should be mass voluntary adherence.”**(Physician)*

#### 3.5.2. Moral Requirement

Many healthcare workers, especially physicians, were in favour of the vaccination requirement, especially in regard to the COVID-19 vaccine. They were in favour of the vaccination requirement, especially to achieve herd immunity as soon as possible and to bring benefits to the whole population. They also thought that it was not preposterous to make vaccination COVID-19 mandatory, especially considering that many vaccinations in Italy already were. Some, however, thought that vaccination for health professionals should be perceived as a moral obligation. They hoped that people would understand the importance of vaccination, regardless of the obligation that could be imposed on them.
*“It’s a moral obligation that health professionals have towards the population. More than an institutional obligation it should be a moral obligation.”**(Physician)*
*“In the past the obligation to vaccinate helped overcome big pandemics and such, so it will with this one! Let’s hope it won’t be needed for people to understand that vaccination is important, even without making it mandatory.”**(Nurse)*

## 4. Discussion

Vaccination attitude is a controversial topic worldwide in different populations, including HCWs [18]. This qualitative study aimed to explore the vaccination attitudes of HCWs in Haematology settings. The results of this study revealed the hidden reasons behind the doubts of HCWs who were hesitant about vaccination and what led some of them to change their opinion. On the other hand, it made it possible to investigate the motivations of those who had a positive attitude towards vaccination. The professionals interviewed were mainly focused on the flu vaccine and the new COVID-19 vaccine. It was possible that participants only addressed the issue of influenza vaccination and no other vaccines (e.g., HPV vaccine, HBV vaccine) because this was the only vaccine that the Italian health system strongly recommended to health workers to get every year, in addition to the new COVID-19 vaccine.

One of the main themes emerged in the interviews was trust. Similar to other findings [11,28], most HCWs showed a high level of confidence in vaccination, while others stated concerns about the long-term effects of the vaccine, the vaccine production process, and little confidence in the pharmaceutical companies. The majority of NAs and nurses stated that they blindly trusted the health authorities responsible for verifying the safety and efficacy of marketed vaccines. The physicians, on the other hand, critically analysed the available studies on the subject to assess the efficacy of the vaccine before having it inoculated. The different form of trust (blind vs. critical) regarding vaccines, between the two professional categories, could be linked to the poor consideration of EBP in nurse aides decision-making [29]. Although trust is a recurring theme in the interviews, it seems that the decision to undergo vaccination is not exclusively linked to this but rather to the personal health choice. The lack of trust did not always imply being entirely opposed to vaccines. Some HCWs, in fact, said that while they maintained scepticism about vaccine efficacy and distrust towards pharmaceutical companies, considering the toll that COVID-19 had already taken worldwide, they were willing to get vaccinated. We had found that confidence, in our context, rather than reusing a true determinant of vaccination, was an indirect variable. In fact, it was often not necessary and/or sufficient in the decision-making process.

We found several reasons behind HCWs’ hesitancy toward vaccination. The most important barrier regarding vaccination was to consider only their individual health in the decision-making process. In this perspective, similar to other findings [30,31,32,33], the perceived lack of benefit of vaccination to themselves and others appeared to play a key role in vaccination refusal. This aspect was more pronounced for the flu vaccine than for the COVID-19 vaccine. It was possible that the higher COVID-19 burden, directly observed by HCWs, could have influenced this occurrence. Some nurses decided not to receive flu vaccination because they did not consider themselves or others susceptible to getting the flu, even in the presence of relevant risk factors (e.g., chronic diseases) of close people (e.g., family members). Additionally, as in another study [29], some of them believed that using standard precautions may be an effective alternative to vaccination. This would merit tailor-made interventions to fill a knowledge gap that may have persisted among some healthcare professionals. Physicians, contrarily, emphasised the patient-protective role that the vaccination of healthcare workers plays. We found that the group consisting of physicians was the most supportive of vaccines, and they tended to have a more community-centred view of health.

One unexpected finding from the interviews was the change in opinion. The triggers that led some participants to decide to vaccinate, despite initially being hesitant, were the opinion of experts and sharing with colleagues. In particular, colleagues seemed to have played a key role in helping an NA understand the importance of vaccination for immunocompromised patients and the community. It could be hypothesised that focusing the minds of vaccine hesitant HCWs on the risks others might face rather than on protecting themselves from the disease might be the key to promoting vaccine uptake. This would be particularly relevant in Haematology departments, where patients are undergoing lymphodepletion treatments, and vaccination is often ineffective for them [9].

The second factor that influenced a change in intentions was the intercession of a persuasive figure. In some cases, the tipping point for the undecided were interviews released by pre-eminent experts in the field of virology and infectious diseases through various media platforms. We also attributed to this category the “lead by example” phenomenon, wherein people felt reassured about the risks and potential side effects once they saw doctors advocating for vaccines, receiving their shots.

A persuasive figure could also be a colleague or friend. In this instance, we found that the opinion of a colleague that was held in high regard was indeed more incisive than the recommendation of the Head of Department or Hospital Management. This indicated that we should acknowledge the responsibility of medical professionals to be informed when discussing health-related topics and presenting their own opinions.

Lastly, we wanted to explore how HCWs felt towards vaccine mandates. At the time the interviews took place, Italian lawmakers were discussing the possibility of requiring all HCWs to be vaccinated against COVID-19. On 1 April 2021, vaccination for HCWs became mandatory for access to the workplace.

Interviewees were split on two stances: those who thought it was necessary to establish mandatory vaccination for COVID-19 for all HCWs and those who dithered on whether that would be an infringement on people’s rights. Many who did not support a state mandate said that they thought it would not be an effective measure to impose vaccine uptake. Some health workers, especially those who had decided to vaccinate for community health, argued that vaccination should be not so much a legal requirement but rather a moral obligation.

## 5. Study Limitations

The study had several limitations. Even if a qualitative approach could explain and deeply understand some complex phenomena such as vaccine hesitancy, it was not possible to characterise the different perspectives between healthcare professions, partly due to the limited number of their representatives. However, the nuances that emerged from the different professionals allowed a deeper description of the categories and a wider framework of the phenomenon. The generalisability of the study findings was limited and was context dependent. Having conducted the interviews over a wide period of time increased the knowledge of the phenomenon studied. However, the interviewees may have been influenced by the contextual and upcoming situations. Although a possible vaccination obligation for healthcare workers was discussed during the first interviews, it had not yet been formalised until 1 April 2021. Therefore, some HCWs may have changed their attitude.

## 6. Conclusions

Non-vaccinated Health Care Workers may be a potential source of infection for patients Compliance with the immunisation schedule of HCWs working in these settings is even more important. The change in opinion of some HCWs interviewed provided insight into the importance of focusing organisational efforts on collective responsibility. Many HCWs felt that concern for their individual health was not a key element in their choice to vaccinate, yet they changed their perspective when they focused on Collective Health. Strategies focusing on collective responsibility could foster the attitude of the more hesitant HCWs. Vaccination communication campaigns focused on the belief that one’s own vaccination will protect others could be a successful strategy to promote a sense of moral duty.

Furthermore, raising awareness among HCWs, especially NAs, about the important risks faced by Haematological patients could encourage their attitude. These communication campaigns should focus on the impact of vaccinations on all preventable diseases for Haematological patients. This also applies to flu vaccinations, which are often not seen as necessary for themselves and others.

## Figures and Tables

**Figure 1 ijerph-20-05901-f001:**
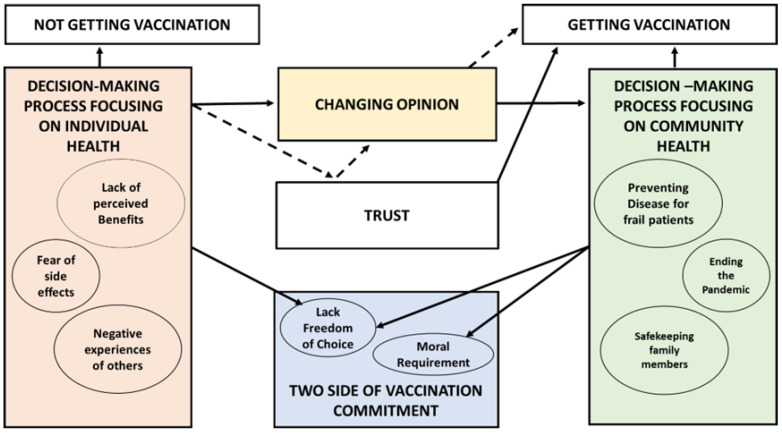
The connection between themes and sub-themes that emerged from the interviews. Arrows are used to explain the relationships between the themes and sub-themes that emerged from the interviews. The continuous arrows (→) represent the direct relationships between the themes/sub-themes, while the discontinuous arrows (- - - >) represent the indirect relationship. The themes are shown by squares, while the respective sub-themes are enclosed within circles.

**Table 1 ijerph-20-05901-t001:** Guiding questions used for the semi-structured interviews.

**Healthcare Professionals Outlook on Vaccination**
- What do you think about vaccination? - Have you ever taken part in a vaccination campaign? - Looking back on the last vaccine you received, how was it offered to you? Was it mandatory? - How was the vaccination campaign carried out in your Institution?
**Questions Specific to SARS-CoV-2**
- What do you think about SARS-CoV-2 vaccination? - What sources of information did you look up regarding vaccines risks and benefits? - Have you ever been approached by acquaintances with questions on vaccines? How did it make you feel? What information did you share with them? How did they react? - Inside your unit, has there been any conversation about vaccination against SARS-CoV-2? - What do you think about the possibility of making vaccination against COVID-19 mandatory? - How did you feel about being one of the first to be vaccinated against COVID-19 in Italy? - During this interview have we touched upon matters you had not considered before? - Is there anything we have not discussed during this interview that you wished to elaborate on?

**Table 2 ijerph-20-05901-t002:** Participants’ demographics.

	N (%)	Median (Min–Max)
Age		47 (26–62)
Gender		
Female	18 (85.7)	
Male	3 (14.3)	
Educational status		
High school	11 (52.4)	
Bachelor’s degree	4 (19)	
Master’s degree	5 (23.8)	
Doctoral degree	1 (4.8)	
Professional category		
Physician	6 (28.6)	
Nurse	7 (33.3)	
Nurse aides	8 (38.1)	
Duration of providing care to patients diagnosed with Haematology disease (years)		17 (1–32)

## Data Availability

The data supporting the presented results of the study are available on request from the corresponding author. The data are not publicly available to protect the privacy of the participants.

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
