# Peer review of "To Get Vaccinated or Not? The Vaccination Decision-Making by Healthcare Professionals Working in Haematology: A Qualitative Study"

_ijerph, 2023, doi:10.3390/ijerph20105901_

Round 1

Reviewer 1 Report

Thank you for your submission. I greatly enjoyed reviewing a qualitative research study and think we get valuable information from these types of studies.

Because of the low number of physicians included in the study, it is questionable whether or not their responses were representative of physicians vs nurses. At some points you clump everyone together as HCW's but then break out physicians verses nurses versus nurse aides. I think it would be helpful to include in your discussion some of the limitations of your study, specifically that the low number of physicians (6) versus nurses (7) versus nurse aides (8) limits the ability to accurately characterize the differences between those 3 groups.

In line 48, the reference should be [9, 10] instead of [910].

In line 52, change "decreased in noso-" to "decrease in noso-".

Overall an excellent article, however, I recommend that the authors include a paragraph discussing the limitations of their study.

Author Response

All the authors would express sincere gratitude for all the suggestions raised by the reviewers, we strongly believe that they have improved the manuscript.  The appreciation expressed is  important to reward our efforts in performing this qualitative research.

Comments

Response

Because of the low number of physicians included in the study, it is questionable whether or not their responses were representative of physicians vs nurses. At some points you clump everyone together as HCW's but then break out physicians verses nurses versus nurse aides. I think it would be helpful to include in your discussion some of the limitations of your study, specifically that the low number of physicians (6) versus nurses (7) versus nurse aides (8) limits the ability to accurately characterize the differences between those 3 groups

Thank you, this is a very interesting point.

In the sampling process and in the data analysis, we have considered all theprofessions together as they all were key informants to generate a description of the vaccine attitude in the HCHs from  different points of view . Nevertheless, as you suggested, no great variance emerged in the points of view of the different HCWs; thus,we just highlighted some minor differences in the categoriesto bring out a wider framework of the phenomenon.

According to your suggestions, we added a specific section on study limitations tomark the low ability to characterize the differences between the different professions..

In line 48, the reference should be [9, 10] instead of [910].

We revised the reference according to your suggestion.

In line 52, change "decreased in noso-" to "decrease in noso-".

We revised the manuscript according to your suggestion.

Overall an excellent article, however, I recommend that the authors include a paragraph discussing the limitations of their study.

Thank you very much for your appreciation.

As you suggested, we added a more extensive paragraph on study limitations.

Reviewer 2 Report

COMPARED TO THE MANY "QUANTITATIVE" RESEARCH ON THE FIELD, THIS "QUALITATIVE" RESEARCH OFFERS US SOME INTERESTING POINTS TO MEDITATE, ESPECIALLY ABOUT MENTAL MECHANISMS THAT BRING HEALTHCARE WORKERS TO VACCINATE OR TO VACCINATE NOT; ABOUT THE MOTIVATIONS OF ACCEPTANCE/REFUSAL, ABOUT THE EGOISTIC/ALTRUISTIC BEHAVIOUR, ETC.

UNFORTUNATELY, THE NUMBERS OF THE REPRESENTATIVES OF THE DIFFERENT CATEGORIES OF HCWS INVOLVED ARE SO LIMITED, THAT IT WAS IMPOSSIBLE TO EVIDENCE THE DIFFERENCES BETWEEN NURSES AND MEDICAL DOCTORS, BETWEEN NURSES GRADUATED IN THE UNIVERSITIES AND THOSE GRADUATED IN REGIONAL SCHOOLS. ALSO A COMPARISON NORTH-SOUTH DOES NOT BRING TO CLEAR CONCLUSIONS.

WHAT MUST BE APPRECIATED IN THIS MANUSCRIPT IS THE ABILITY TO BRING TO OUR CONSIDERATION THE MENTAL MECHANISMS OF THE HCWS TOWARD ACCEPTANCE OR REFUSAL, AND THE RESULTS IN SHOWING US THE EXTREME VARIABILITY OF WHAT STAYS IN THE MIND OF HCWS ABOUT VACCINES AND VACCINATION. WE MUST APPRECIATE THAT THE RESEARCHERS WERE ABLE TO MAKE IT EVIDENT, IN OUR HCWS, THE FREQUENT PRESENCE OF ALTRUISTIC (COLLECTIVE) CONSIDERATION FOR "THE OTHERS" AND THE AVAILABILITY TO CHANGE THEIR MIND AFTER DISCUSSION WITH OTHER HCWS AND ACQUISITION OF THE OPINION OF RESPECTED AND RECOGNIZED LEADERS. I AM NOT SO SURE THAT THIS BEHAVIOUR IS SO POPULAR AMONG A LOT OF HCWS!

MINOR REVISIONS ABOUT THE ENGLISH LANGUAGE:

LINE 41 (HAS SHOWED)

LINE 46 ("EFFICACY" SHOULD BECOME "EFFECTIVENESS"

LINES 60-61 (the delay in acceptance or to refusal of vaccination) LINE 73: (it is enough to write "HCWs", because the explanation has ben already given at line 54)

Author Response

All the authors would express sincere gratitude for all the suggestions raised by the reviewers, we strongly believe that they have improved the manuscript.  The appreciation expressed is  important to reward our efforts in performing this qualitative research.

Comments

Response

Unfortunately, the numbers of the representatives of the different categories of hcws involved are so limited, that it was impossible to evidence the differences between nurses and medical doctors, between nurses graduated in the universities and those graduated in regional schools. Also a comparison north-south does not bring to clear conclusions.

Thank you for raising this interesting point.

In the sampling process and in the data analysis, we have considered all the professions together as they all were key informants to generate a description of the vaccine attitude in the HCHs from  different points of view . Nevertheless, as you suggested, no great variance emerged in the points of view of the different HCWs; thus,we just highlighted some minor differences in the categories. Moreover, we added a specific section on study limitations to highlight this issue.

Actually, we have not made a comparison between Northern and Southern contexts, but we have realized that Table 1 was confusing. Therefore, we have embedded the two columns in one to make the information more consistent with the performed analysis and to avoid possible misunderstandings in the results section.

line 41 (has showed)

line 46 ("efficacy" should become "effectiveness"

We changed the text according to your suggestion.

LINES 60-61 (the delay in acceptance or to refusal of vaccination)

LINE 73: (it is enough to write "HCWs", because the explanation has ben already given at line 54)

We revised it according to your suggestion.